# Position: Lifetime tuning is incompatible with continual reinforcement learning

**Golnaz Mesbahi** [1 2]  **Parham Mohammad Panahi** [1 2]  **Olya Mastikhina** [1 2]  **Steven Tang** [1 2]  **Martha White** [1 2 3]
**Adam White** [1 2 3]

## Abstract

In continual reinforcement learning (RL) we want agents capable of never-ending learning, and yet our evaluation methodologies do not reflect this. The standard practice in RL is to assume unfettered access to the deployment environment for the full lifetime of the agent. For example, agent designers select the best performing hyperparameters in Atari by testing each for 200 million frames and then reporting results on 200 million frames. In this position paper, we argue and demonstrate the pitfalls of this inappropriate empirical methodology: *lifetime tuning*. We provide empirical evidence to support our position by testing DQN and SAC across several continuing and non-stationary environments with two main findings: (1) lifetime tuning does not allow us to identify algorithms that work well for continual learning—all algorithms equally succeed; (2) recently developed continual RL algorithms outperform standard non-continual algorithms when tuning is limited to a fraction of the agent's lifetime. The goal of this paper is to provide an explanation for why recent progress in continual RL has been mixed and motivate the development of empirical practices that better match the goals of continual RL.

## 1. Do not peek at the test set!

Continual reinforcement learning (RL) arises in many applications. In HVAC control, agents learn to adapt process setpoints daily, with deployment lasting for weeks or months, but the agent does not exploit knowledge of the length of the deployment (Luo et al., 2022) and the agent designer might not know how long the agent will run. Similar situations arise in data-center cooling (Lazic et al., 2018), water treat-

ment (Janjua et al., 2024), and many other industrial control settings. Even our popular deep RL benchmarks could naturally be treated as continual RL tasks: Atari agents could play games forever, switching to a new game when they die or complete each game (similar to the Switching ALE benchmark (Abbas et al., 2023)). Mujoco tasks are naturally continuing, but common practice is to truncate experiments after a fixed number of interactions, resetting to some initial configuration. In continual RL tasks, we should design and evaluate our agents with limited access to the environment and then deploy the learning system as-is without further tuning of its hyperparameters during the rest of its lifetime.

The vast majority of algorithmic progress in deep RL has focused on the non-continual setting. Agent designers test algorithmic variations and hyperparameter combinations in the deployment environment for the full lifetime of the agent and then report the best performance across these deployments. For example, if one were to develop a new exploration algorithm for Atari, then this new algorithm would be extensively tested over 200 million frames, tuning any new hyperparameters introduced by evaluating each over 200 million frames. In this sense, the standard methodology is to design and evaluate our agents given access to the full, finite, lifetime of the agent. This is critical because it means that designers can tweak hyperparameters such that performance is optimized for a particular lifetime length. For example, one could tune the epsilon decay schedule of DQN to maximize area under the learning curve, generating the best result possible over 200 million frames. But if that agent was run for 800 million frames, or 10 million frames, the resultant performance would be suboptimal.

There has been increased focus on extending or modifying existing deep RL agents for continual RL, with limited success. These approaches can be roughly categorized into three groups: 1) resetting, 2) regularization, and 3) normalization. In the first approach, parts of the agent's network are reset to random initial values, causing large drops in performance but eventually leading to improved final performance (Nikishin et al., 2022; 2023; D'Oro et al., 2022). Regularization balances error reduction with keeping the agent's network parameters close to initialization (Kumar et al., 2024); this helps because the random initial parameters help the network learn quickly. Finally, recent work

[1]Department of Computing Science, University of Alberta, Edmonton, Canada [2]Alberta Machine Intelligence Institute (Amii) [3]Canada CIFAR AI Chair. Correspondence to: Adam White <amw8@ualberta.ca>.

*Proceedings of the 42nd International Conference on Machine Learning*, Vancouver, Canada. PMLR 267, 2025. Copyright 2025 by the author(s).

has found that layer normalization can help maintain the ability to learn (Lyle et al., 2023). All these approaches are mitigations: algorithmic fixes applied to a base agent that is not designed for continual RL. In all these works, the empirical demonstrations were conducted in non-continual testbeds like Atari and Mujoco, where the proposed new continual learning agents were tuned for the agent's entire lifetime—not a continual learning setting.

Unfortunately, lifetime tuning is particularly misleading for continual learning research. Many recent papers follow the same basic template: (1) introduce a continual variant of an existing benchmark (typically by introducing a source of non-stationarity); (2) demonstrate that existing algorithms fail on the new benchmark; (3) introduce a new algorithmic mitigation and demonstrate that it solves the new continual benchmark. There are multiple ways this can be misleading. It is possible that in step two, simply retuning an existing algorithm's hyperparameters could eliminate the failure (fair and statistically sound treatment of hyperparameters is rare, see Patterson et al. (2024b;a); Jordan et al. (2024)). There is no reason to expect default hyperparameters to work well in a new, non-stationary task. A more subtle issue, is that in step 3, the new continual learning algorithm was likely tuned over the lifetime of the experiment. That means we do not know if the algorithm will work if the lifetime is longer; we do not know if the agent can learn continually!

This paper argues the position that **progress in continual RL research has been held back by inappropriate empirical methodologies, specifically *lifetime tuning***. We discuss an alternative methodology for tuning and evaluating continual RL agents inspired by the constraints of real-world applications of RL. One approach is based on a simple idea: continual RL agents may be deployed for an unknown amount of time and thus agent designers should not be allowed to tune their agents for their entire lifetime. Instead, we suggest a limited tuning phase: a small percent of the total lifetime. Only $k$-percent of the experiment data can be used for hyperparameter tuning; after that, the hyperparameters must be fixed and deployed for the remainder of the agent's lifetime. This proposal is not meant to replicate a real-deployment scenario. Instead, the idea is to constrain agent evaluation in a way that would better highlight the benefits of algorithms (1) with fewer task-specific hyperparameters, (2) that use meta-learning to adapt their own hyperparameters automatically, and (3) that can learn continually. The goal of arguing this position and our simple proposed evaluation methodology is to encourage the development of agents that are more suitable for continual RL and perhaps deployment in the real world, not introduce a way to tune hyperparameters.

We provide a collection of experiments to support our position. We show that a widely-used deep RL agent, DQN, performs poorly across a suite of continual RL tasks despite testing different metrics to select the best hyperparameters under $k$-percent tuning. We additionally test Soft Actor-Critic, to show the impact of $k$-percent and lifetime tuning on a different algorithm in a continuous action setting, finding similar outcomes. We also show that the value of $k$, the interaction budget for tuning, that achieves good lifetime performance can be agent-environment dependent. We also investigate several mitigation strategies, which do not appear beneficial under lifetime tuning, and show they actually improve performance compared to the base algorithms under $k$-percent tuning.

## 2. Background and Problem Formulation

We consider continual RL problems formulated as Markov Decision Processes (MDPs) with partial observability. On each discrete time step, $t = 1, 2, 3, ...$, from the current state $S_t \in \mathcal{S}$, the agent selects an action $A_t \in \mathcal{A}$ and the environment transitions to a new state $S_{t+1} \in \mathcal{S}$ and emits a scalar reward $R_{t+1} \in \mathbb{R}$. The agent may only observe a partial view of this state, $x_t \doteq x(s_t)$. A continual RL problem is one with a long agent-environment interaction—either one long episode as in a continuing problem or many episodes as in an episodic problem[1]—that is eventually truncated at an unknown time $T$. Neither the agent nor the agent designer can exploit this information because it is unknown. The agent essentially needs to treat this $T$ as infinite, even though we evaluate it for a finite time.

There are several possible formal definitions of continual RL (Abel et al., 2023; Khetarpal et al., 2022; Sutton et al., 2007), but for the purposes of this paper, the continual RL problem is one where there is no fixed optimal policy and thus the agent must explore, learn, and adapt its behavior forever. Such continual or neverending learning is needed in (vast) environments with long lifetimes and partial observability or nonstationarity. These environment properties are also inherent in continual RL applications, such as robotics or industrial control, with long agent-environment interaction and inherent partial observability from limited sensors.

In our experiments, partial observability is typically artificially introduced, usually by adding some source of nonstationarity to an existing non-continual, stationary environment. For example, in Non-stationary Mountain Car the force of gravity changes slowly over time according to some unobservable schedule. The main idea is that because the

---

[1]Episodic problems are ones where the agent-environment interaction naturally breaks up into sub-sequences where the agent reaches a terminal and then is teleported to a start state. A continuing problem formulation has no such termination. In our experiments, we use transition-based discounting to unify the treatment of episodic and continuing problems; see White (2017) for further details.

environment is changing forever, the agent must continually adapt forever. This kind of setup is a stand-in or emulation of the idea that there are always new things to learn about, just like in our own lives. We will consider such non-stationary environments in this work, and run demonstrative experiments with DQN (Mnih et al., 2015) for discrete actions and Soft Actor-Critic (SAC) (Haarnoja et al., 2018) for continuous actions.

## 3. Lifetime tuning in continual RL

Hyperparameters have a dramatic impact on both the performance and learning dynamics of deep RL agents. DQN is one of the simplest such agents and it contains 16 hyperparameters (see Mnih et al. (2015) for details) controlling size of the replay buffer, target network updated rate, averaging constants in the Adam optimizer and exploration over time, to name a few (Adkins et al., 2024). These hyperparameters allow us to instantiate variants of DQN that learn incredibly slowly to mitigate noise and off-policy instability, to fast online learners that can track non-stationary targets (for example, see Huang et al. (2022)).The proliferation of hyperparameters in modern Deep RL agents effectively allows the agent designer to select which algorithm they want to use ahead of time for a given problem. This is even more important in continual RL, as recent work has shown that the default hyperparameter settings of popular agents must be significantly adjusted to deal with long-running non-stationary learning tasks (Lyle et al., 2023).

Let us define lifetime tuning in the context of conventional non-continual, stationary RL. Imagine an environment, such as ALE (Bellemare et al., 2013; Machado et al., 2018), where the length of the agent-environment interaction (the lifetime) is fixed and known to you the agent designer. If you were to develop a new algorithm for ALE, let's call it DQN++, you would periodically test your latest, greatest version on several Atari games, each with a 200 million frame lifetime. You would run the experiment multiple times, likely testing DQN++ with different hyperparameter settings, all for 200 million frames. Over the entire development cycle of DQN++ you have designed, tuned, and fit your new method to this particular 200 million frame lifetime. You may have overfit to both the environment (see Whiteson et al. (2011)) and the lifetime you have chosen.

Tuning agents for a particular lifetime becomes problematic in the context of continual RL. Recall it is typical to create new continual RL benchmarks by adding a source of non-stationarity. One view of non-stationarity is that there are aspects of the MDP that are not fully observable to the agent, thus the dynamics appear non-stationary. This is easy to see because the vast majority of benchmarks in RL are computer programs with well-defined internal state and state-transition mechanisms. Our non-stationary, continual learning problems simply do not make parts of the internal state or transition mechanism visible to the agent.

If we allow lifetime tuning in a non-stationary environment, then we compromise the utility of these non-stationary benchmarks for testing continual learning algorithms. We inadvertently overcome the partially observability by iteratively designing, tuning, and testing our agents for the entire lifetime. Every run of the experiment reveals more about the hidden dynamics to the researcher and the learning algorithm. The more we run, the more we reveal. In the end, the benchmark is not nearly as partially observable (non-stationary) and no continual learning is actually needed. We compromise the original goal of these continual (partially observable) testbeds, which was to force us to build agents that could adapt to the unexpected.

## 4. A hypothetical continual RL experiment

We illustrate this point with a sequence of toy experiments: a hypothetical progression that a researcher might follow. Perhaps she starts with a simple, conventional benchmark, like the Catch environment famously used to develop the DQN algorithm. In Catch, the observation and input to the agent is a bitmap image. Every episode a ball drops from the top of the screen and the agent's goal is to catch the ball with a paddle at the bottom, that the agent can move left, right, or stay. The agent either catches (+1 reward) or misses the ball (-1 reward) and then the episode ends and a new ball is randomly spawned at the top of the screen. As we can see in Figure 1 (lhs), a DQN agent can perform well on this environment, learning quickly and achieving stable performance at the end of the lifetime. We will be more rigorous about presenting experimental details in the next section, here the intention is to focus on the big picture.

Our researcher next develops a continual variant of Catch, to further demonstrate DQN is not well suited for continual RL. This variant of Catch does not have an episodic structure. Instead, on every step, a new ball will spawn randomly at the top of the screen with some small probability (0.1), meaning now there can be multiple balls at once. The respawn probability is set such a that an agent, on expectation, can catch the ball. To make the environment non-stationary, we select two pixels in the observation and swap them every 10,000 steps. The network must continually relearn the meaning of pixels in the observation. As we see in Figure 1(rhs) DQN indeed fails on this new environment.

In the next step, the researcher develops a new continual learning algorithm to tackle this non-stationary benchmark. Perhaps she explores using regularization to keep the network parameters close to their initial values (Kumar et al., 2024), called W0-DQN. Our researcher is very likely to implicitly over-fit W0-DQN to Non-stationary Catch through

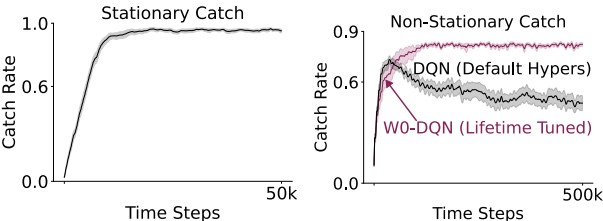

*Figure 1.* DQN performs well **(lhs)** in the simple stationary catch environment, but never reaches the same performance when applied to non-stationary Catch **(rhs)**, even if we run the experiment 10 times longer. In fact, DQN gets worse with time, the signature of loss of plasticity (Dohare et al., 2021; Lyle et al., 2022).

repeated testing and hyperparameter tuning, a process sometimes called *designer bias* (Patterson et al., 2024b).

Consider how this would unfold in Non-stationary Catch. Over a fixed lifetime, the agent will see a fixed number of pixel swaps. Our researcher can make the buffer size of W0-DQN smaller to quickly purge the inaccurate transitions after each switch. It does not matter whether our researcher discovers this by exploiting knowledge of the problem or directly tuning the hyperparameters (including buffersize) for performance. Even if the switch rate changed with time also, performance tuning over a finite lifetime would give a large advantage. As we see in Figure 1 (rhs), if we are allowed to systematically sweep W0-DQN's hyperparameters over the full lifetime and report the best performance, then we see no loss of plasticity.

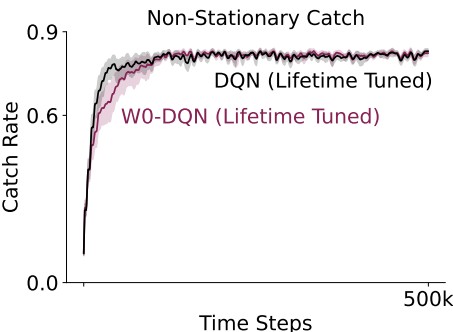

*Figure 2.* Both DQN and W0-DQN perform nearly identically in a non-stationary Catch if we allow lifetime tuning.

However, if we also apply lifetime tuning to DQN, even in this non-stationary environment, we see good performance as highlighted in Figure 2. We simply used a grid search and lifetime tuning. This makes sense because there is no reason to expect that the hyperparameters that work well in Catch should work well in Non-stationary Catch—DQN was significantly disadvantaged because it was untuned. This is an important step that is often missed! Our researcher might

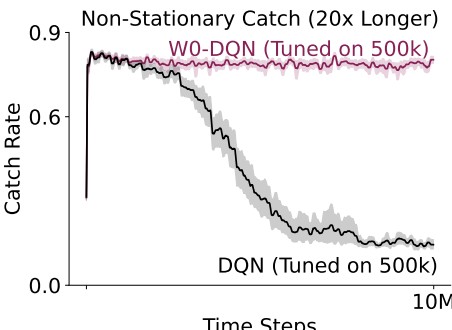

*Figure 3.* W0-DQN can significantly outperform DQN in non-stationary Catch if we run the experiment 20 times longer. Here we use the same hyperparameters previously found to be best in non-stationary Catch, tuned over a 500k lifetime.

have convinced herself that DQN was hopeless and that W0-DQN is significantly better in continuing environments based on the results in Figure 1 (rhs). And the conclusion would make sense to her and be less likely to be doubted, as DQN was designed for stationary environments, whereas regularizing the weights in this way has been previously shown to be effective in continual RL.

In summary, there are two potential pitfalls here. (1) We may falsely believe an algorithm cannot perform well in continual RL if its hyperparameters are not appropriately adjusted for a new environment. Continual learning papers often introduce new environments, making this pitfall likely. (2) If we do lifetime tune algorithms, then we might not conclude an algorithm designed for continual learning is actually better, because all algorithms perform similarly under lifetime tuning. To see why, let's run one more experiment.

A reasonable continual learning agent should be able to continue to learn, even if you run your experiment longer and longer. Let's take our two agents from the previous experiment, now reasonably tuned for our new Non-stationary Catch environment—that is, not using some community established default hyperparameters—and run our experiment 20 times longer. Figure 3 clearly shows a significant difference in performance between the two methods: vanilla DQN's performance collapses whereas W0-DQN performs well for the entire duration of the experiment. W0-DQN performs well without tuning its hyperparameters for the entire lifetime—it is better at continual learning, in this environment at least. If we were to retune both agent's hyperparameters for this much longer lifetime, we might end up back in the situation summarized in Figure 2 and fail to show the obvious benefit of W0-DQN in this non-stationary, continual learning environment.

# 5. One alternative: $k$-percent tuning

Although, there are many possible alternatives to lifetime tuning, we propose a very simple one here as a starting place, called $k$-percent tuning. The name describes the relatively simple idea: we propose to tune the agent only for $k$-percent of its lifetime. We as experimenters will usually know how long we will run our experiment for, but at least we can constrain ourselves to tune only over a small window. If the agent will run for $n$ steps, then we tune the agent for $j = \lfloor kn \rfloor$ steps, where $k$ is a percentage in $[0, 100]$. In other words, for every hyperparameter setting suggested via grid search or Bayesian optimization (Parker-Holder et al., 2022; Eimer et al., 2023), we run the agent for $j$ steps and record the agent's performance. We then chose the best hyperparameter configuration, according to the performance metric of interest (e.g., total return over the tuning phase). The agent is then deployed with these hyperparameters for the full $n$ steps, for multiple runs, to get the performance over the full lifetime of the agent.

# 6. The impact of $k$-percent tuning in small-scale experiments

In this section, we explore how $k$-percent tuning impacts performance in a simple continual stationary RL environment. Continuing Cartpole (Barto et al., 1983) is a simple classic control environment with completely stationary dynamics. The agent's observations are the position and velocity of the cart and the angle and angular velocity of the pole. The actions are discrete, move left or right, the discount is 0.999, and the goal is to keep the pole balanced, without hitting the ends of the track. The reward is $+1$ for every step that the pole is balanced. Once the pole falls more than 24 degrees from its upright position, the agent receives a reward of 0, and the pole is reset to the upright position, but the agent is not reset. We plot an exponential moving average (0.99 averaging constant) of the reward.

We consider a large set of hyperparameters for DQN, sweeping exploration (epsilon), batch size, buffer size, minimum buffer size, and the values of learning rate and $\beta_2$ of the Adam optimizer. The ranges and chosen hyperparameters listed in Tables 1 and 2 respectively. [2]

We tested three different criteria for selecting the best hyperparameter configuration during the tuning phase. In particular we tried: (1) performance over the last 10% of the tuning

---

2Although it is usually the case that larger learning rates are chosen over shorter tuning windows in k%-tuning, we found it does not always happen and is problem dependent. Over a short period of time Cartpole is easy and many hyperparameter settings nearly tie in performance: AUC of 0.9904788 for $\alpha = 0.01$ vs 0.9887988 for $\alpha = 0.1$. This is so close that the best hyperparameters chosen by the sweep are subject to stochasticity. Note, k%-tuning did select one of the larger learning rates, but not the largest.

| Learning rate | $10^i : i \in [-1, \cdots, -5], 0.08$ |
| Batch size | 32, 256 |
| Buffer size | 1,000, 10,000, 100,000 |
| Min buffer size | 0, 1000 |
| Exploration $\epsilon$ | 0.01, 0.1 |
| Adam optimizer $\beta_2$ | 0.9, 0.999 |

*Table 1.* DQN hyperparameter ranges tested.

phase, (2) total cumulative performance over tuning (i.e., area under the curve or AUC), and (3) the hyperparameters that resulted in the best performance when looking at the worst performing seed. In our experiments all three criteria produced similar results, so we report AUC for simplicity.

| | $k\%$-tuned | lifetime tuned |
|---|---|---|
| Learning rate | 0.01 | 0.08 |
| Batch size | 256 | 256 |
| Buffer size | 1,000 | 1,000 |
| Min buffer size | 0 | 0 |
| $\epsilon$ | 0.01 | 0.1 |
| Adam $\beta_2$ | 0.999 | 0.9 |

*Table 2.* Best performing hyperparameters for DQN on Cartpole.

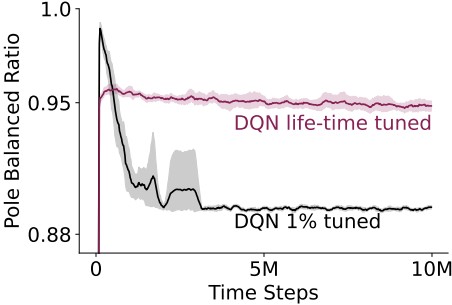

*Figure 4.* Tuning on one-percent of a lifetime leads to poor performance in Continuing Cartpole, whereas lifetime tuning allows DQN to achieve nearly optimal performance. Results are averaged over ten independent trials and the shaded regions are 95% bootstrap confidence intervals.

In Figure 4 we again see the impact of lifetime tuning. If we are only allowed to tune DQN's hyperparameters for a fraction of the lifetime, the performance is high early in learning and then catastrophically collapses. Our results on Non-stationary Catch and Cartpole, two simple toy environments, suggests DQN is not well suited for continual learning. DQN's performance under lifetime tuning potentially suggests that some prior success of recent continual RL algorithms might be explained by lifetime tuning.

# 7. Recent continual learning algorithms are less reliant on lifetime tuning

There have been numerous mitigation strategies introduced in the continual RL literature, designed to make algorithms like DQN more robust. In this section, we show that some of these mitigations actually work well under $k$-percent tuning.

We investigated several recent mitigation strategies.

**W0Regularization(W0)** (Kumar et al., 2024): The distance between the current and initial weights is added to the loss to encourage the weights to stay near the initialization.

**L2Regularization(L2)** (Dohare et al., 2024; van Laarhoven, 2017): A term proportional to the $\ell_2$ norm of the weights of the network is added to the loss function to ensure the weight magnitudes are kept small.

**CReLU** (Abbas et al., 2023): The concatenated ReLU activation function limits the number of inactive units by concatenating the output of ReLU($x$) with ReLU($-x$). This should reduce the percentage of dead neurons since CReLU maintains 50% of the neurons in an active state.

**PT-DQN** (Anand & Precup, 2023): The value function is decomposed into two separate networks: permanent and transient. The transient is updated toward the residual error from combining both networks' predictions and is reset periodically. The permanent network is only updated by distilling the transient network's predictions.

**Weight normalization** (Salimans & Kingma, 2016): Weight matrices are split into the weight magnitudes and weight directions, with separate gradients for each.

**Layer Normalization** (Ba et al., 2016): This method applies normalization to activations of the neural network by using the statistics from all of the summed inputs to the neurons within one layer.

**$k$-percent tuning of DQN mitigations:** Figure 5 summarizes the performance of DQN with mitigations under one-percent tuning in Non-stationary Catch and Continuing Cartpole. All mitigations except Layer Normalization perform well in Non-stationary Catch. In Continuing Cartpole, performance is more mixed. DQN with Layer Normalization outperforms lifetime-tuned DQN, whereas PT-DQN more or less matches lifetime-tuned DQN. The other mitigations, perform worse, but none are as bad as DQN (in black). This is significant because in Figure 5 all the algorithms have their hyperparameters set using one-percent tuning.

This result suggests that lifetime tuning can make new algorithms appear less useful because their performance does not surpass an unreasonable standard set by lifetime tuning.

**$k$-percent tuning of SAC mitigations:** We ran a similar experiment with SAC in a modified environment from the

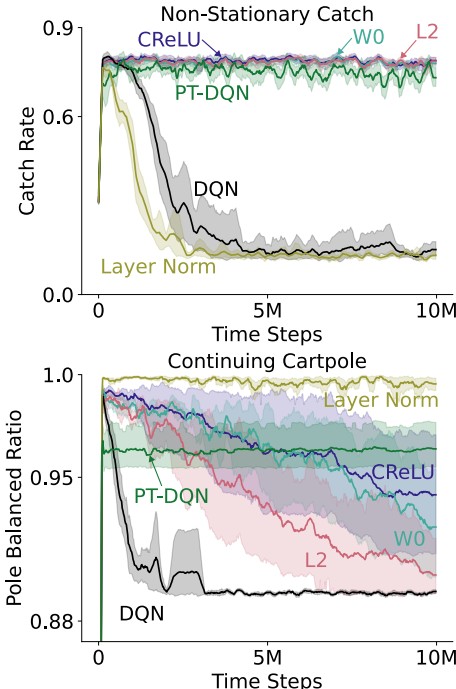

*Figure 5.* The performance of several mitigation strategies designed to improve the performance of DQN in continual learning environments. All results plotted correspond to the best performing hyperparameters under one-percent tuning. Results are averaged over ten independent trials and shaded regions show the 95% bootstrap confidence intervals.

DeepMind Control Suite (Tassa et al., 2018). We investigated how SAC performs with one-percent tuning in a lifelong learning setting where the environment switches from quadruped-walk to quadruped-run halfway through the experiment. We again considered a large set of hyperparameters for SAC, including the learning rate, batch size, buffer size, and the values of $\beta_2$ and $\epsilon$ in the Adam optimizer summarized in Table 3. The ranges and chosen hyperparameters are outlined in Table 4. We compare the one-percent-tuned values with the default hyperparameters previously reported for the DeepMind Control Suite (Haarnoja et al., 2018).

| Learning rate | $2 \cdot 10^{-2}, 10^i : i \in [-2, \cdots, -7]$ |
|---|---|
| Batch size | 16, 32, 128, 256, 512 |
| Buffer size | 512, 1000, 10000 |
| Adam optimizer $\beta_2$ | 0.9, 0.999 |
| Adam optimizer $\epsilon$ | $1 \cdot 10^{-8}, 0.1$ |

*Table 3.* Hyperparameter ranges for one-percent tuning on SAC on DeepMind Control Suite environments

Figure 6 shows the performance of SAC with different mitigations under one-percent tuning in the switching Quadruped-walk-run environment. Most mitigation strate-

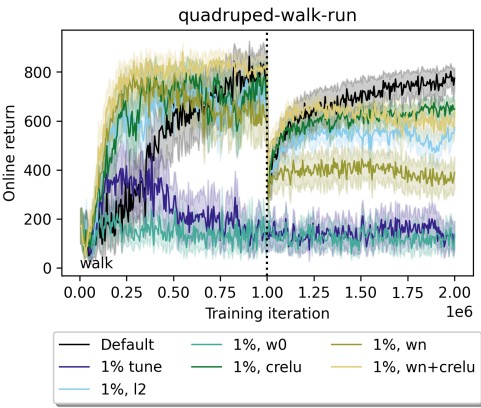

*Figure 6.* Multiple mitigation strategies do improve the performance of quadruped-walk-to-run with hyperparameters obtained from tuning on one-percent of quadruped-walk. $l2$ is weight decay $= 1 \cdot 10^{-5}$, $w0$ is with penalization of weights moving away from their initialization values, and $wn$ is weight normalization. The results are based on 10 runs, and the shading is the standard error.

gies improve performance over SAC with one-percent tuning, except for W0Regularization which further decreases performance. CReLU improves performance the most on its own, and combining CReLU with weight normalization has the strongest effect. Interestingly, weight normalization on its own is the least effective when moving from walk to run. Note, in Table 4 the learning rate chosen by one-percent tuning in quadruped-walk-run is larger than the default. The learning rate is particularly sensitive to lifetime. In shorter lifespans, tuning selects for larger learning rates that are beneficial in the short run, but $k$-percent tuning highlights how this can result in poor performance in the long run.

Normalization has been shown to allow for larger learning rates (Bjorck et al., 2018; Salimans & Kingma, 2016; Ba et al., 2016) and that may be why weight normalization works well in Quadruped-walk-run. Although $\ell_2$ regularization has been shown to increase the effective learning rate (van Laarhoven, 2017), it is not sufficient here.

| | default | $k\%$-tuned |
|---|---|---|
| Learning rate | $3 \cdot 10^{-4}$ | $1 \cdot 10^{-3}$ |
| Batch size | 256 | 512 |
| Buffer size | $1,000,000$ | $10,000$ |
| Adam optimizer $\beta_2$ | 0.999 | 0.9 |
| Adam optimizer $\epsilon$ | $1 \cdot 10^{-8}$ | $1 \cdot 10^{-8}$ |

*Table 4.* Default hyperparameter values and those selected using one-percent tuning for SAC on the DeepMind Control Suite. Tuning was done with three independent runs.

## 8. $k$-percent tuning in a never-ending task

In this section, we contrast using $k$-percent tuning and lifetime tuning to compare continual learning mitigation strategies for DQN in Jelly Bean World, a testbed for never-ending, continual learning (Platanios et al., 2020). We use the environment configuration detailed in (Anand & Precup, 2023), where the agent navigates through up, down, left, right actions, and has to adapt to a non-stationary reward function that flips between +2 and -1 for collecting jelly-beans and onions respectively, every 0.15 M steps. The discount factor is 0.9. We compare lifetime tuning where hyperparameters are tuned on the full 1.5 M steps, with 10% and 20% tuning, where hyperparameters are tuned using only 0.15 M and 0.3 M steps respectively. With 10% tuning, the agent is tuned for the time steps before the first reward function flip, so the environment appears stationary. Under 20% tuning, both reward functions are observed by the agent, providing information regarding the nature of the non-stationarity in the environment.

We evaluate two mitigation strategies applied to the DQN algorithm: W0Regularization (W0-DQN) and permanent transient networks (PT-DQN). We follow a two-stage tuning approach selecting hyperparameters using 5 independent trials and then evaluating them using 30 new independent trials (Patterson et al., 2024b). We tuned DQN with a fine-grained sweep over 25 learning rates, $\alpha \in \{10^{-12/6}, 10^{-13/6}, \ldots, 10^{-36/6}\}$ so a similar amount of computational resources is used for hyperparameter tuning between methods. For W0-DQN, we tune the learning rate $\alpha \in \{10^{-2}, 10^{-3}, \ldots, 10^{-6}\}$ and the regularization coefficient $\lambda \in \{10^{-2}, 10^{-3}, \ldots, 10^{-6}\}$. For PT-DQN, we tune the permanent value function learning rate $\alpha_P \in \{10^{-4}, 10^{-5}, \ldots, 10^{-8}\}$ and the transient value function learning rate $\alpha_T \in \{10^{-2}, 10^{-3}, \ldots, 10^{-6}\}$. All other hyperparameters are set to the values used in prior work (Anand & Precup, 2023).

Figure 7 shows a comparison between lifetime tuning and $k$-percent tuning for DQN, W0-DQN, and PT-DQN in Jelly Bean World. There were no significant differences between all three algorithms with lifetime tuning, except in the last few reward function swaps (near the end) DQN takes longer to recover than the variants with mitigations. However, with 20% tuning, W0-DQN and PT-DQN performed well in this non-stationary environment, while DQN performance declined over time with 20% tuning. With 10% tuning, DQN and PT-DQN's performance collapses over time, while W0-DQN performance was maintained. The best performing algorithms (e.g., W0-DQN) do not exhibit loss of plasticity, but do exhibit a saw-tooth pattern of interference and re-learning because the networks are not recurrent and cannot remember previous tasks.

The choice of lifetime or $k$-percent tuning has a significant

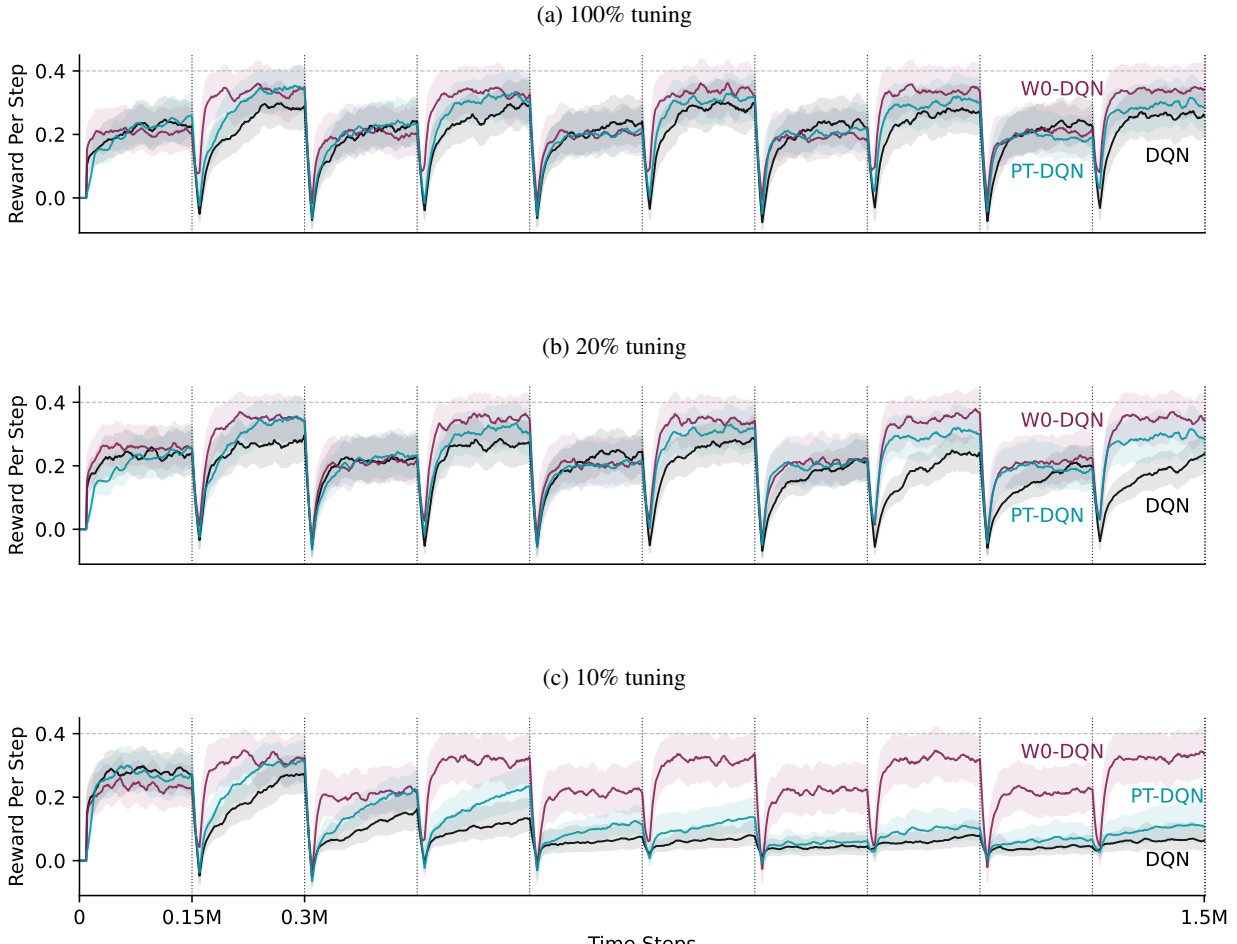

*Figure 7.* W0-DQN and PT-DQN perform well in **Jelly Bean World**, even when restricted to 10% and 20% tuning respectively. All algorithms, including DQN, perform well when using lifetime tuning. Results are averaged over 30 independent trials and the shaded regions are 95% bootstrap confidence intervals.

impact on the conclusions one can draw in Jelly Bean World. DQN with $k$-percent tuning exhibits loss of plasticity and no longer learns. The mitigations in W0-DQN and PT-DQN with 20% tuning adapt to this non-stationary environment. However, for PT-DQN there is a large performance drop when using only 10% tuning, likely because the agent only observes one reward function during tuning and is unable to adapt to a challenging lifetime of switching between the two reward functions. In contrast, W0-DQN is more robust under 10% tuning, as it is able to maintain performance without observing the reward function non-stationarity during the tuning phase. However, if we only look at the lifetime tuning results, we cannot draw any conclusions on the effectiveness of the mitigations.

The results in Figure 7 clearly demonstrate how lifetime-tuning can both obfuscate good performance and inflate performance. Looking at the top row (100%) of this figure, one might conclude that all three agents perform similarly

in JBW. However, as the amount of tuning decreases, we see DQN's performance is clearly decreasing with time. At the most extreme, 10% tuning, it becomes very clear that W0-DQN significantly outperforms the other methods and that PT-DQN performs no better than DQN.

## 9. Alternative Views

An obvious alternative position is the null position: we do not need new methodologies, we just need better algorithms. It is true that algorithm development is not always hampered by current empirical practices nor lack of problem formulations. In RL, however, there is historical precedent of the significant positive impact of standardization that is well-aligned with exploiting current algorithm's limitations. Take, for example, the Arcade Learning Environment (ALE) (Bellemare et al., 2013; Machado et al., 2018). Before ALE was introduced, little empirical work in RL was applicable to multiple environments and image-based observations.

These limitations were known, but the benchmarks and evaluation methodologies of the day were not suitable for such ambitious experiments. In particular, the original ALE paper was very particular in its methodological suggestions (e.g., tuning on five games and reporting online performance). We cannot know how long progress would have been delayed without these concrete proposals. The position of this paper is that we, are again, in need of new methodology and benchmarks, this time for continual RL.

It is commonly held that, in deep RL, the community has settled on a common set of default hyperparameters that work well across environments for popular agents and thus tuning is no longer a confounding factor. There is significant evidence supporting this position. The well-loved 37 implementation details of PPO outlines choices that have allowed many researchers to get PPO working on a variety of environments. The leaderboard results on Spinning Up Baselines show that SAC is still near SOTA, with one set of hyperparameters, across several Mujoco tasks—these results were compiled in 2018! More recently, the DreamerV3 architecture achieved high performance, with one set of hyperparameters, across dozens of environments, including Minecraft (Hafner et al., 2025).

On the other hand, there is also evidence to the contrary. Recent work has shown that PPO and innovations introduced in DreamerV3 actually induce high sensitivity (Adkins et al., 2024), the amount of data required for statistically sound rankings of algorithms is much larger than standard practice (Jordan et al., 2024; Patterson et al., 2024a), and the empirical practices used to identify community defaults (i.e., common practice) may be suspect (Patterson et al., 2024b). Unfortunately, our algorithms are still very sensitive to the choice of hyperparameters. We do not know how much additional performance we forgo with defaults and a new environment could cause existing methods (with default choices) to utterly fail. We do not yet have general agents, thus (lifetime) tuning matters.

AutoRL (Parker-Holder et al., 2022; Eimer et al., 2023) and, more generally, meta-learning approaches for setting agent hyperparameters may appear to side-step the issue of how tuning is conducted. Generally speaking, these methods use an outer meta-learning process to adapt the hyperparameters of the underlying RL algorithms—to learn how to learn. Methods have been developed to adapt the step-size parameter (Dabney & Barto, 2012; Jacobsen et al., 2019), the eligibility trace parameter (White & White, 2016; Mann et al., 2016; Kobayashi, 2022), and even the update itself (Xu et al., 2018; Flennerhag et al., 2022). These meta approaches, like the underlying algorithms, have hyperparameters as well, such as how long each hyperparameter combination is evaluated, how initial hyperparameter's are sampled, exploration strategy in the hyperparameter-space,

etc. In fact, meta approaches have been largely overlooked in RL in favor of default hyperparameter settings discovered via lifetime tuning. An online meta-learning approach that continually adapts agent hyperparameters could tune during the whole lifetime of the agent and thus more clearly highlight the benefits of meta learning. We believe meta-learning approaches appear less useful because they are compared with life-time tuning.

One might easily conclude from our exploration of $k$-percent tuning that this approach does not help find good hyperparameters for continual learning and thus is not useful. It is certainly true that several of our results show suboptimal performance, failure, and even loss of plasticity. On the other hand, we also saw that some algorithms designed for continual RL outperform baselines and even do well. Lifetime tuning makes algorithms perform similarly, whereas $k$-percent tuning highlights the promise of these methods in a more challenging empirical setup.

Related, $k$-percent tuning is not really similar to a real deployment scenario and thus using it will not directly tell us which algorithms will perform well in applications. This is certainly true. Although our position against lifetime tuning was based on, in part, the argument that lifetime tuning is not possible in reality, $k$-percent tuning is limited in this respect. Yes it is hard to imagine a deployment scenario where we have limited access to a system, but we can run multiple independent trials (seeds). One could certainly propose an evaluation methodology more useful for deployment: something like $k$-percent with a single trial. However, our purpose was to provide a simple alternative to generate results that highlight the problems with lifetime tuning. This is a position paper with a call to action: stop lifetime tuning your continual RL agents. We hope and expect future work to improve upon and replace $k$-percent tuning.

We cannot prevent people from cheating in their evaluations. This is a valid criticism. Furthermore, we cannot prevent people from cheating accidentally, just due to lack of knowledge. This lack of knowledge is exactly the motivation for arguing this position and inviting the community to establish better evaluation methodologies for continual RL. If there is a well-specified alternative to lifetime tuning, then people can adopt it into their workflows. If we want to prevent misleading evaluations, then that may require establishing environment servers that researchers connect with to run their experiments, as used on the RL competitions (Whiteson et al., 2010). For now, we think that is a step too far, and $k$-percent tuning is a small step in the right direction.

## Impact Statement

This paper presents work whose goal is to advance the field of Machine Learning. There are many potential societal

consequences of our work, none which we feel must be specifically highlighted here.

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
