# OpenReview forum: "Position: Lifetime tuning is incompatible with continual reinforcement learning"
_ICML.cc/2025/Position_Paper_Track — ICML 2025 Position Paper Track poster_

### Official Review · Reviewer_vxxu · 2025-03-08

**Significance:** 4
**Argument Clarity:** 3
**Rating:** 4
**Confidence:** 3

**Questions:**

1. Can the authors more concretely and specifically define the continual reinforcement learning problem in Section 2? The one currently provided seems a bit ad-hoc.
2. For k-percent tuning, how might researchers select the value of k? This seems to be another hyperparameter that might not be standardized for specific environments, and could lead to results that are difficult to compare (at different levels of k).

**Discussion Potential:**

2

**Paper Summary:**

The paper considers the setting of continual reinforcement learning, where in contrast to “traditional” episodic reinforcement learning, the agent is required to learn a policy for an infinitely long, continuous episode without resets. The paper argues that a major issue in current research evaluation methodology for continual learning methods is that they often use hyperparameters that are tuned by looking at the performance across the entire lifetime (for some fixed length of time). This is a process that is unrealistic to perform in a real world continual learning setting, and the authors show empirical examples that demonstrate that this can lead to misleading research conclusions about the effectiveness of or lack of effectiveness of continual reinforcement learning algorithms. The authors advocate instead for a strategy called k% tuning, in which the algorithm hyperparameters are tuned on a fraction of the total episode length, rather than the entire lifetime.

## update after rebuttal

I appreciate the authors' clarifications and maintain my recommendation for acceptance.

**Position:**

Yes

**Position In Title:**

Yes

**Related Work:**

3

**Strengths And Weaknesses:**

Strengths:
- The topic is relevant and important to the community. I believe there is a growing interest in continuous (reinforcement) learning settings, and this paper presents an insightful and nuanced argument for researchers in this field.
- The position is well supported by evidence. I think the didactic example and empirical evidence presented in Section 4 are very informative and help to identify the crux of the problem identified in the position of this paper.
- The suggestion of k% tuning is also very sensible, and I think if adopted by researchers, it will help to streamline the progress in this research direction. Even if it is not widely adopted, I think it will be nice to spur additional discussion.

Weaknesses:
- I think the main weakness of this paper is its presentation – I think the main arguments could be formulated in a more rigorous and clean way. For instance, Section 1 seems to be a bit of a hybrid introduction/related work, and much of the exposition before the position is presented doesn’t directly relate to or motivate the stated position.
- There are also some typos and grammatical errors, e.g. in the title of Section 1 “peek” is spelled incorrectly (unless I am missing a pun). Additionally, some phrasing is a bit too conversational “the idea that there are always new things to learn about, just like in our own lives,” “If you were to develop a new algorithm for ALE, let’s call it DQN++, you would periodically test your latest, greatest version on several Atari games, each with a 200 million frame lifetime.” While I appreciate the effort to make the text straightforward and approachable, this exposition seems more fitting for a blog post than a paper.
- Additionally, the title of the paper doesn’t obviously correspond to the main position presented in this paper to me. After reading the paper, I wouldn’t agree that the main argument is that lifetime tuning is incompatible with continual RL, which implies that one somehow cannot perform lifetime tuning on continual RL. The position, as stated in Section 1, is that it is a poor empirical methodology.


In summary, I believe this work identifies a practical and important problem in the research methodology (evaluation) in continual reinforcement learning. It provides empirical evidence of the issue in an approachable way, and then presents a call to action with a proposed solution. While this work may not generate a huge amount of discussion (since I think the position it is advocating for seems very reasonable given the experimental evidence), I do think that it is an important consideration for researchers in the area.

**Support:**

3

---

> ### Author Rebuttal · Authors · 2025-03-28
>
> Thank you for your insightful comments and detailed feedback!
>
> # Questions:
>
> **1. The definition of continual reinforcement learning**
>
> You are right that we spread the definition of continual RL across multiple paragraphs, instead of using a crisp mathematical definition in one paragraph. Because this is a position paper and not a technical paper, we intentionally avoided introducing notation that we would not need to use. Additionally, we started with the standard MDP definition, which readers are comfortable with, and then explained the key part of continual RL (the one long interaction of unknown length T).
>
> However, in retrospect, we can see how this might actually be harder to follow, and feels adhoc. We could change this to: (1) introduce partial observability upfront, when introducing the MDP, stating that the agent observes a potentially partial view of s_t, x_t= x(s_t). Then (2) immediately state that the agent has one long environment interaction of length T. Then, in the second paragraph, we could justify how this matches existing definitions and could explain how it encompasses both continuing and episodic environments [1].
>
> [1] Abel, D., Barreto, A., Van Roy, B., Precup, D., van Hasselt, H. P., & Singh, S. (2023). A definition of continual reinforcement learning. Advances in Neural Information Processing Systems, 36, 50377-50407.
>
> **2. For k-percent tuning, how might researchers select the value of k? This seems to be another hyperparameter that might not be standardized for specific environments, and could lead to results that are difficult to compare (at different levels of k).**
>
> This is a difficult question. We do not see $k$ as a hyperparameter that researchers should be tuning for performance; but as you note, people will! The expensive answer is researchers could report results with several different values of $k$. Perhaps the most reasonable answer is for designers of new partially obs, continual RL benchmarks to specify a value of $k$ they think is reasonable. That establishes the practice and is used for the first results with the new benchmark. Naturally, researchers that follow may decide to change $k$ because it better matches their research question: perhaps $k$ extremely small because they are interested in studying and highlighting the merits of parameter free algorithms (for example, algorithms based on coin betting [2]). As long as researchers are clear about their research question, what $k$ they used and why, we think it might be okay.
>
> Another (more far out) option is to take inspiration from real-world problems where researchers might only have access to their deployment setting for a small testing window. After that they need to deploy an algorithm! The choice of $k$ could be inspired by the ratios of (1) testing on the system vs (2) deployment periods, that are common in real-world settings
>
> [2] Jacobsen, A., & Chan, A. (2021). Parameter-free gradient temporal difference learning. arXiv preprint arXiv:2105.04129.
>
> # Comments:
>
> **1. Section 1 seems to be a bit of a hybrid introduction/related work**
>
> Correct, despite the name, Section 1 is indeed an “Introduction” section. We followed the style of including related work in the introduction. We will revise the text to make it more clear and clean.
>
> **2. Conversational text**
>
> We will revisit and revise.
>
> **3. Suggestion on the title**
>
> We agree; the focus is poor methodology. We will revise.

---

> > ### Comment · Reviewer_vxxu · 2025-04-03
> >
> > Thank you for the response and clarifications. I don't agree that position papers and technical papers are mutually exclusive (my understanding is that position papers are also meant for the same technical audience as main track papers). But, it's fair to not introduce unused notation, I think it'd be best to pick whichever option allows the text to be most precise and clear. I trust the authors will revise the other parts of the text, and I maintain my rating and opinion that this is a solid paper.

---

### Official Review · Reviewer_w878 · 2025-03-11

**Significance:** 3
**Argument Clarity:** 3
**Rating:** 4
**Confidence:** 4

**Questions:**

1. One issue I see with $k$-percent tuning is that it seems senstive to the absolute value of the lifetime. I think we see this with some of your results, where $k$-percent tuning currently tends to lead to overly high learning rates. Do you think one part of the solution should maybe, in combination with $k$-percent tuning, simply be to run evaluations for much longer? Or would this put us back into overfitting territory again? I imagine that tuning hyperparams on only 10% of the lifetime if the lifetime is only 1M steps can lead to too high of a learning rate. But what if our evaluation lifetime is 100M steps, instead of just 1M? Wouldn't hyperparams tuned on 10% of that much larger lifetime already start to translate much more effectively?

**Discussion Potential:**

3

**Paper Summary:**

This paper describes *lifetime tuning* as the (currently common) practice of tuning hyperparameters of Deep RL algorithms using the same number of time steps that will subsequently also be used for evaluating the final algorithms (the algorithms with tuned hyperparams). The paper's core position is that this practice should not be used for research on Continual RL, as in reality, we won't know in advance for how long we'd be deploying (or we would be deploying forever). One suggestion for an improved methodology is $k$-percent tuning, in which only $k$ percent of the time steps that will ultimately be used for final evaluations are used for hyperparam tuning. Various experiments show the impact that the different methodologies can have on the results and the conclusions that may be drawn.

## update after rebuttal

After rebuttal and reading other reviews, I maintain my "Accept" score.

**Position:**

Yes

**Position In Title:**

Yes

**Related Work:**

3

**Strengths And Weaknesses:**

### Strengths

- Clear position, well-motivated, good experiments demonstrating the relevance of paying attention to the issues raised. Likely to inspire discussion in the Continual RL community.

### Weaknesses

- I don't know if "lifetime tuning" is already a well-established phrase (I think not?), but I find it somewhat confusing. Well, once I see what you mean with it, I see it, and then it makes sense. But when I first saw the phrase, my immediate interpretation of it was "tuning (hyperparameters) automatically throughout the entire lifetime of an agent". That sounds to me like something that would actually be a very good thing to do in Continual RL (with the question of how to do it well of course being a very challenging question). But you mean something very different with it. If you prefer to keep it as is, I think it'd be okay, but I think I would suggest something slightly different. Maybe something like "lifetime-aware tuning" or "horizon-aware tuning"?
- The terminology/notation around $k$-percent tuning is imprecise. Percentages range from $0$ to $100$, not from $0$ to $1$. When you currently write that you do $k$-percent tuning with $k = 0.1$, you mean that you use $10\%$ of the number of time steps, but most of your text actually implies that this would be using only $0.1\%$ of the number of time steps. Percentages and ratios are not the same thing, there's a $100\times$ difference between them.
- The second paragraph of Section 8 describes tuning learning rate for DQN, and learning rate + one other hyperparam for each of W0-DQN and PT-DQN. The same number of different values is tested for learning rate for each algorithm. But DQN only tunes one hyperparam, whereas the others tune over a 2D grid of 2 hyperparams. This is unfair. To make it fair, you could have tested  more different values for the learning rate of DQN (i.e., use the same amount of computation resources to tune learning rate for DQN in a more fine-grained manner than you do for the other algs, as the other algs have a second hyperparam to be tuned).

### Other comments

- Please cite the reference for "The well-loved 37 implementation details of PPO"

**Support:**

4

---

> ### Author Rebuttal · Authors · 2025-03-28
>
> Thank you for your insightful comments and detailed feedback!
>
> # Questions:
>
> **1. “One issue I see with k-percent tuning is that it seems sensitive to the absolute value of the lifetime.”**
>
> This is a fascinating question! Running the tuning longer could certainly help—it would expose more of the data distribution the agent is likely to encounter over its lifetime. There are at least three issues with that. (1) It makes our experiments longer and more expensive—k% is very simple, but one of its clear merits is that it makes things run faster. (2) In some situations, say in the real world, we might have limited interactions with the environment for tuning purposes and so we can’t just run longer. If we care about that setting, then we might worry about the run longer approach. (3) Finally, in principle, we could be interested in continual problems that truly do require unending learning and 10 M, 100 M, 1 trillion steps all really do require different hypers—at least with current popular algorithms. This is why meta-learning approaches are fascinating because they could continuously tune over the whole experiment, thus side stepping these issues—assuming the hyper-hyperparameters of the meta learning algorithm are not sensitive. Or just better algorithms that have fewer and less sensitive hypers. Lifetime tuning obfuscates how useful such algorithms could be in this regard!
>
> # Suggestions:
>
> **1. "lifetime-aware tuning"**
>
> Brilliant suggestion! We will use it!
>
> **2. Please cite the reference for "The well-loved 37 implementation details of PPO"**
>
> Will do!
>
> **3. The terminology/notation around k-percent tuning is imprecise**
>
> We will clean this up for camera ready.
>
> **4. Use the same amount of computation resources to tune learning rate for DQN in Section 8**
>
> We will test this. If the result is different, we will include both in the paper (one in the appendix, one in the main text), otherwise, we will include a footnote that we tried this and the result was not different. We note that in this experiment lifetime-tuned DQN does well compared with k%-tuned DQN, and we expect this change will improve the performance of both variants. Thus the conclusion regarding how these two compare should remain the same---which supports the main position of the paper.

---

### Official Review · Reviewer_Vaeq · 2025-03-13

**Significance:** 3
**Argument Clarity:** 4
**Rating:** 4
**Confidence:** 4

**Questions:**

**Questions**

See weaknesses.

1. Why does the title of the first section mention “test set” when there is no mention about test set anywhere in paper? Moreover, there is no train-test split in continual RL which is causing more confusion.
2. What are the 14 hyperparameters in DQN? Can the authors include them in a footnote or the appendix for completeness?
3. In the catch game, how do you ensure that the agent can continue catching all the balls? That is, how do you ensure that the agent has sufficient time to catch two consecutive balls when they appear in extreme ends?
4. Typically, when part of the agent’s lifetime is chosen for selecting the hyperparameters, algorithms end up selecting aggressive values to accrue more returns (eg. higher learning rate). However, in Table 2, the learning rate value for the lifetime tuned agent is higher than the k-percent tuned. Why?

**Areas of improvement**

1. The final paragraph in the introduction (contributions) can be presented as bullet points to increase readability.
2. “[...] for the purposes of this paper, [...]” this line can benefit from citing [1].
3. Although k-percent tuning supports the stated position, it is still useful to list the downsides of the the proposed approach for tuning hyperparameters. For example, the designer doesn’t know how many steps exist in the agent’s lifetime or what’s the best k to choose.
4. It would be useful to highlight (a) how prevalent “lifetime tuning” is in continual RL by citing and highlighting the practice in the literature; (b) papers where k-percent tuning in different forms already exist (eg. PT-DQN uses 1.5M timesteps to select HPs while the results are reported for 2.1M).

**References**

[1] Sutton, Richard S., Anna Koop, and David Silver. "On the role of tracking in stationary environments." Proceedings of the 24th international conference on Machine learning. 2007.

**Discussion Potential:**

3

**Paper Summary:**

The paper takes the position that tuning hyperparameters on the entire experience of the agent—a prevalent approach for tuning hyperparameters in continual reinforcement learning—is incorrect. The stated position is supported by twofold empirical evidence: (a) “lifetime tuned” non-continual RL algorithms (eg. DQN) perform equally well compared to continual RL algorithms (eg. PT-DQN), which can lead to incorrect conclusions regarding the continual learning ability of non-continual RL algorithms; (b) several continual RL strategies fail when the tuning is limited to k-percent of the agent’s experience, which highlights the ineffectiveness and frailness of several existing continual RL strategies. The paper ends with a call for action to the continual RL community to develop novel methods for hyperparameter selections that don’t use the full agent’s experience.

**Position:**

Yes

**Position In Title:**

Yes

**Related Work:**

3

**Strengths And Weaknesses:**

**Strengths**

- The draft is well-written, easy to read, and organized well; especially the way it builds up from small-scale to large-scale experiments in a nice flow.
- Although a trivial strategy, k-percent tuning is simple and effective to the extent that it supports the paper's position. It also serves as a good baseline to compare against for any future, complex approaches for hyperparameter selection.
- The discussion on hyperparameter selection for continual RL is timely and much needed as there’s an increase in interest in this topic in the field.
- The results presented in Figure 5 are striking—the fact that many commonly used mitigation strategies fail on simple problems further highlights the importance of having proper hyperparameter selection methods in continual RL.

**Weaknesses**

- When papers report results by tuning hyperparameters on the entire experience of the agent, they are not arguing for “lifetime tuning”. Instead, the results should be interpreted as: if the best hyperparameters can be found for each method at every point in time, then algorithm A performs better than algorithm B. From this perspective, “lifetime tuning” is not being performed in the current research.
- The paper has little to no discussion on “meta” approaches for hyperparameter tuning. Since “meta” approaches continuously tune hyperparameters, they are “lifetime tuners”. And, such methods are desirable for continual RL as the agent needs to change its hyperparameters based on the changes in the environment (eg. more exploration is needed when the agent steps into previously unexplored parts of the environment).
- Although some proposed algorithms use “lifetime tuning” for hyperparameter selection, the selected hyperparameter values are transferable to many new environments without needing further tuning. For example, DQN uses the same set of hyperparameters on all the ALE games. Likewise, Dreamer uses the same hyperparameter values across several domains.

**Support:**

4

---

> ### Author Rebuttal · Authors · 2025-03-28
>
> Thank you for you comments and suggestions!
>
> # Questions:
> **1. Why “test set”?**
>
> The point of this name was to emphasize how lifetime tuning in continual RL is similar to making use of the test set in supervised learning (SL). Another name for this section could simply be “Introduction” as that is what that section really is. We will adjust this for camera ready.
>
> **2. What are the 14 hyperparameters in DQN?**
>
> This number comes from a recent study of hyperparameters in RL [1]. It looks like [1] was updated slightly, and there are 16 hyperparameters. We will update our text, include the citation and provide the list of 16 hypers in the appendix.
>
> [1] https://proceedings.neurips.cc/paper_files/paper/2024/file/e1cadf5f02cc524b59c208728c73f91c-Paper-Conference.pdf
>
> **3. [In Catch] How do you ensure that the agent can continue catching all the balls?**
>
> The board has 10 rows and 5 columns. The agent must move horizontally to catch the falling ball. If a ball spawns at one extreme end (at the top), the agent has 10 time steps to cross at most 5 spaces. This ensures that it has sufficient time to react and reach the ball. This was introduced in Deepminds csuite (https://github.com/google-deepmind/csuite). We will add this as a footnote.
>
> **4. In Table 2, the learning rate value for the lifetime tuned agent is higher than the k-percent tuned. Why?**
>
> Although it is usually the case that larger learning rates are chosen over the short tuning window, we found it does not always happen and is problem dependent. Over a short period of time cartpole is easy and many hyperparameter settings nearly tie in performance: AUC of 0.9904788 for alpha=0.01 vs 0.9887988 for alpha=0.1. This is so close that the best hypers chosen by the sweep are subject to stochasticity. Note, k% did select one of the larger learning rates, just not the largest. We will add a footnote about this for camera ready. In our experience with continuing cartpole, agents suffer significant loss of plasticity over the full lifetime. This could explain why lifetime-tuned agents prefer larger alpha and larger exploration epsilon.
>
> # Comments:
>
> **1. “lifetime tuning” is not being performed in the current research**
>
> As you point out, if some process could be used to find the best hypers for every lifetime, then comparison claims would stand. We agree! Our conjecture is that current algorithms would require re-tuning for many different lifetimes. Furthermore, because current empirical practices do not highlight this, some algorithms will outshine other algorithms that are actually more robust in the continual RL setting. We like this point and will try to adjust the text to reflect this nice way of putting it.
>
> **2. More discussion of “meta” approaches for hyperparameter tuning.**
>
> We agree these meta approaches are critical! In fact, we believe lifetime tuning effectively obfuscates the benefits of meta approaches in continual RL. We discuss AutoRL in the last section of the paper, but we will add additional discussion of meta approaches!
>
> **3. Hyperparameter transfer across domains**
>
> It is true that some hyperparameter values remain stable across domains, there are also several recent papers showing that hyperparameters are not transferable in different domains [1, 2, 3]. DQN is a great example of this: the hyperparameter settings that work in ALE, exhibit no learning in some other environments! DreamerV3 is certainly an interesting case where one hyper-set worked for many environments (though not well in Acrobot). Although, a recent paper showed some of the ideas introduced in DV3 do induce hyper sensitivity when combined with PPO [1]. I think it is fair to say: we do not yet have parameter free (insensitive) learners in RL. In continual RL, things appear to be different! It is possible that some of the reported failure of continual RL could be due to poor transfer of hypers, as we tried to illustrate in section 4. A recent paper showed that retuning, in such cases, can mitigate loss of plasticity [4].
>
> [2] https://arxiv.org/abs/2406.17523v3
> [3] http://arxiv.org/abs/2407.18840
> [4] https://proceedings.mlr.press/v202/lyle23b/lyle23b.pdf
>
> **4. Please highlight (1) how prevalent “lifetime tuning” is in continual RL and (2) papers things like k-percent are used.**
>
> We have already done this lit review, but the discussion of it  in Section 1 can be improved, as you note. As you mention, the PTW paper uses something like k%. Interestingly, something similar to k% has been used continual SL: restricting access to a subset of tasks [5]. We will include this citation and any others we can find in the camera ready version.
>
> [5] https://openreview.net/forum?id=Hkf2_sC5FX
>
> # Suggestions:
>
> **1. Cite “On the role of tracking in stationary environments”**
>
> Great point. Will do
>
> **2. Downsides of k%**
>
> We tried to outline this in the last section (#8) of the paper. We will expand the text in this regard.

---

### Official Review · Reviewer_AY8j · 2025-03-16

**Significance:** 2
**Argument Clarity:** 3
**Rating:** 2
**Confidence:** 3

**Questions:**

Catastrophic forgetting is mentioned rather later in the paper, but shouldn't this be introduced  earlier in the paper?
Isn't PPO also a way to  keep policy close to before?

**Discussion Potential:**

2

**Paper Summary:**

This paper points out the limitation of conventional training and evaluation practice in RL and considers ways to address continual learning problems.

**Position:**

No

**Position In Title:**

No

**Related Work:**

2

**Strengths And Weaknesses:**

This paper deals with a concrete problem, but it is more like a technical paper rather than a position paper.

**Support:**

2

---

> ### Author Rebuttal · Authors · 2025-03-28
>
> Thank you for your comments.
>
> # Questions:
> **1. “Isn't PPO also a way to keep policy close to before?”**
>
> Yes, PPO is designed to stay close to the current policy. However, this is determined by the setting of hyperparameters, and recent work has shown that PPO is sensitive to the setting of these hyperparameters [1]. In addition, it is widely held that PPO is stable and has good default hyperparameters for popular benchmarks. However, we do not have strong evidence that this is the case in continual RL environments.
>
> [1] Adkins, J., Bowling, M., & White, A. (2024). A Method for Evaluating Hyperparameter Sensitivity in Reinforcement Learning. Advances in Neural Information Processing Systems, 37, 124820–124842.
>
> **2. "Catastrophic forgetting is mentioned rather later in the paper, but shouldn't this be introduced earlier in the paper?"**
>
> As you rightly point out we don't even use this term in the paper, we talk about "interference" in Section 8. In camera ready we can introduce "Catastrophic forgetting" in the background section. Note, in continual reinforcement learning both catastrophic forgetting and loss of plasticity occur together. Forgetting was not really the focus of our position or the paper, but something we observed in the final experiment.
>
> # Comments:
> **1. The reviewer suggested the paper was not a position paper (technical) and that the position was not in the title.**
>
> Naturally, this is a matter of opinion and we respect the reviewer’s point of view. The other reviewers all agreed this was a position paper. Please note, empirical results were included in the paper in order to support the position about empirical practices in continual RL. We proposed a simple improvement to current practice as a call to action and demonstration of what is possible.

---

> > ### Comment · Reviewer_AY8j · 2025-04-05
> >
> > I could understand the meaning of this paper by reading other reviewers' comments, and revised my scores. I also think lifetime tuning is not a good expression.

---

### Decision · Program_Chairs · 2025-04-26

**Decision:**

Accept (poster)

**Comment:**

Stengths: well-written, clear position, position well supported by evidence, understandable alternative, crystal clear call to action.
Weaknesses: limited discussion of meta approaches (that potentially side-step concern of tuning), some confusion about the position being taken, the call-to-action approach is also potentially sensitive to the percentage of an agent's lifetime that tuning is done over, lifetime tuning is an ambiguous term.

Overall, there was broad consensus that the paper's position is timely and well supported. Noted weaknesses are suggestions for improvement but not critical blockers to acceptance. Several of the weaknesses were pre-addressed by the authors in the submission and the main ask for a revision would be to read the reviews and consider if the remaining confusion that led to these concerns can be addressed.